# Effect of Extraction Time on the Extractability and Physicochemical Properties of Pepsin—Soluble Collagen (PCS) from the Skin of Silver Catfish (*Pangasius* sp.)

**DOI:** 10.3390/gels9040300

**Published:** 2023-04-03

**Authors:** Mannur Ismail Shaik, Intan Nordiana Md Nor, Norizah Mhd Sarbon

**Affiliations:** Faculty of Fisheries and Food Science, Universiti Malaysia Terengganu, Kuala Nerus 21030, Malaysia

**Keywords:** collagen, yield, extraction time, physicochemical properties, silver catfish

## Abstract

The current study aimed to determine the effects of extraction time on the extractability and physicochemical properties of collagen from the skin of silver catfish (*Pangasius* sp.). Pepsin soluble collagen (PSC) was extracted for 24 and 48 h and analysed in terms of chemical composition, solubility, functional group, microstructure, and rheological properties. The yields of PSC at 24 h and 48 h extraction time were 23.64% and 26.43%, respectively. The chemical composition exhibited significant differences, with PSC extracted at 24 h showing better moisture, protein, fat, and ash content. Both collagen extractions indicated the highest solubility at pH 5. In addition, both collagen extractions exhibited Amide A, I, II, and III as fingerprint regions for collagen structure. The morphology of the extracted collagen appeared porous with a fibril structure. The dynamic viscoelastic measurements of complex viscosity (η*) and loss tangent (tan δ) decreased as temperature increased, and the viscosity increased exponentially as the frequency increased, whereas the loss tangent decreased. In conclusion, PSC extracted at 24 h showed similar extractability to that extracted at 48 h but with a better chemical composition and shorter extraction time. Therefore, 24 h is the best extraction time for PSC from silver catfish skin.

## 1. Introduction

Collagen is a vital animal protein, constituting around 30% of total protein, making it the most abundant fibrous protein in animals [1,2]. It is essential in forming connective tissues such as bones, tendons, skin, and the vascular system [3,4]. Collagen’s biocompatibility, biodegradability, and weak antigenicity make it a valuable resource in the food, cosmetic, and pharmaceutical industries, as well as in medical treatments for various diseases [5]. However, collagen is extracted from the skin of land-based bovine and porcine animals and marine sources such as the skin, bones, scales, and fins of various marine animals [6,7]. Collagen derived from marine sources is precious, due to the prevalence of numerous diseases in land animals, including bovine spongiform encephalopathy (BSE), avian flu, and foot-and-mouth disease (FMD). Additionally, various forms of land animal collagen are prohibited for Muslims, Jews, and Hindus [8,9]. Studies on collagen extracted from marine sources, such as the skin of Sepia pharaonis [10], fringescale sardinella (*Sardinella fimbriata*) waste [11], sharpnose stingray [7], and silver catfish (*Pangasius* sp.) skin [12] have reported promising results as an alternate for the land animal sources.

Four methods are commonly used to extract collagen: the salting-out method, the alkali treatment, the acid treatment, and the enzyme method [13]. Usually, collagen is extracted using acid treatment (acid-soluble collagen, ASC) or enzymatic methods (pepsin-soluble collagen, PSC). Acid extraction may use organic or inorganic acids; however, the organic acid is more effective than inorganic acid [4,14]. The use of enzymes to extract collagen protein has been an ideal method recently due to its superior reaction selectivity and the fact that it is less destructive to collagen protein [13]. Pepsin, trypsin, ficin, pancreatin, bromelain, and papain are the enzymes used to extract collagen protein. Pepsin is commonly used in enzyme extraction due to its higher yields and better collagen properties.

The success of collagen extraction depends on various parameters, such as the concentration of acid and enzymes, the temperature of extraction, pH, concentration, and time. Extraction time is one of the factors that contribute to successful collagen extraction. Differences in the yields of collagen extracted from different fish species and extraction times have been reported, such as for *Clarias* species (a hybrid species of *Clarias gariepinus* × *C. macrocephalus*), pangasius catfish (*Pangasius sutchi*), black tilapia (*Oreochromis mossambicus*), and sultan fish (*Leptobarbus hoevenii*), with extraction times ranging from 4 to 12 h and yields varying from 299.93 to 368.36 mg/g for PSC and 97.52 to 139.71 mg/g for ASC [15]. Extraction can be conducted using different acids (HCl, acetic acid, and lactic acid) at different times (24, 48, and 72 h). Acetic acid or lactic acid gives maximal yields of about 90%, while HCl offers yields of about 18%, as measured by Skierka and Sadowska [16].

Collagen extraction time does not only influence yield. In addition, it may affect the extracted collagen’s physicochemical properties such as molecular weight, thermal stability, and mechanical properties. The properties of marine collagen have been reported to be similar to those of mammalian collagen. Chemical characterization of collagen includes its chemical composition, amino acids, and pH determination. The physical characterization of collagen includes molecular and thermal stability, functional groups, microstructure, solubility, and rheological properties. For example, Ahmad et al. reported that collagen-extracted PSC from unicorn leather jackets for 48 h showed the highest solubility at pH 2 [6]. However, a study by Huang et al. [17] found that PSC extracted from the balloon fish skin for 24 h showed the highest solubility at pH 5.

Silver catfish (*Pangasius* sp.) is one of Malaysia’s most popularly consumed freshwater fishes [12,18]. Silver catfish are generally known as sutchi catfish or iridescent shark catfish. In Malaysia, silver catfish are known as “patin.” Other names worldwide include “pa sooai” in Laos, “plasawai” in Thailand, and “catra” in Vietnam [19]. This species is fast-growing and is thus commonly cultured in ponds, pens, and floating cages [18]. Several studies have been conducted using silver catfish, such as the determination of fatty acid composition [20], the isolation and characterization of acid (ASC) and pepsin (PSC) soluble collagen [12,21], and the physicochemical properties of skin gelatin derived from silver catfish (*Pangasius sutchi*) as affected by the extraction time [18]. As a result, the current study’s objectives were to extract silver catfish (*Pangasius* sp.) skin collagen at various extraction times and to characterize the physicochemical properties of each sample extracted.

## 2. Results and Discussion

### 2.1. Yield of Extraction

Table 1 shows the yield of PSC extracted from the skin of silver catfish (*Pangasius* sp.) at 24 and 48 h of extraction time. There was no significant difference in yields between 24 and 48 h of extraction. The similar extractability of PSC at 24 and 48 h extraction times was due to the degree of hydrolysis of the enzyme pepsin [1]. In most cases, the intermolecular crosslinks, which increase the solubility of collagen molecules, often occur in the telopeptide region and are aided by pepsin [22]. The crosslinked peptides (PSC at 24 and 48 h) formed a covalent bond via the condensation of aldehyde groups at the telopeptide region. Therefore, these cross-linked molecules can be cleaved without damaging the triple helix’s integrity [23]. The use of the skin of silver catfish (*Pangasius* sp.) as the collagen source extracted for 24 h and 48 h extraction times led to the same amount of collagen yield, probably due to the enzyme’s optimal level, which can maximize the extracted collagen yield depending on factors such as the nature of the enzyme and the raw samples that were used [24].

A study conducted on grass carp (*Ctenopharyngodon idella*) skin by Zhang et al. [25] found that a 12 h extraction time resulted in a higher yield (14.8%) compared to a 24 h extraction time, which contrasts with the finding of the current study. However, collagen yield remained constant when the extraction time was extended to 72 h. A similar finding was supported by Peck and Mashitah’s [15] study on Malaysian freshwater fish muscles: *Clarias* species (a hybrid of *C. gariepinus* × *C. macrocephalus*), pangasius catfish (*Pangasius sutchi*), black tilapia (*Oreochromis mossambicus*), and sultan fish (*Leptobarbus hoevenii*), which showed an increased yield in PSC from 1 to 20 h of extraction time. However, further prolongation of the extraction time after 20 h showed no further improvement in yield.

### 2.2. Chemical Composition of Extracted Collagen

Table 2 presents the chemical composition of PSC extracted from the skin of silver catfish (*Pangasius* sp.) at 24 and 48 h of extraction times. The chemical composition for PSC at 24 h and 48 h extraction time was 93.15 ± 0.48% and 89.93 ± 0.35% for moisture content, 53.22 ± 2.34% and 45.18 ± 1.24% for crude protein, 2.95 ± 0.31% and 2.73 ± 0.81% for crude fat, and 21.86 ± 0.88% and 26.69 ± 1.680% for ash content, respectively. There was a significant difference in the chemical composition between PSC at 24 h and 48 h extraction time (*p* < 0.05). This may be due to the degree of hydrolysis [13] and that, in turn, might be because PSC at 24 h extraction time had completely hydrolysed the non-helix peptide chain of collagen protein, which led to the highest percentage of chemical composition with a high purity of collagen. However, after the 24 h extraction time, the ability to hydrolyse remained at the same level [26].

The moisture content becomes the main composition part on PSC at both the 24 h and 48 h extraction times, in which both collagen samples possessed a moisture content of 93.15% and 89.93%, respectively. This finding was supported by studies conducted by Hukmi and Sarbon [12] on PSC extracted for 30 h from silver catfish (*Pangasius* sp.) skin, which showed 88.25% of moisture. Both collagen extractions have a high amount of ash with 21.86% and 26.69%, respectively, for PSC at the 24 h and 48 h extraction times. The high ash content of extracted PSC at the 24 h and 48 h extraction times might be due to the absence of a demineralization process before the extraction process. According to Matmaroh et al. [27], after demineralization, about 98% of inorganic matter will be discarded. Fat content for PSC at 24 h and 48 h extraction time showed low fat content, which was 2.95% and 2.73%, respectively. Low fat content on PSC at 24 h and 48 h extraction time might be due to the successful defatting process before the extraction method. However, the results obtained in this study did not agree with the study conducted by Kittiphattanabawon et al. [28], who stated that protein content has a higher value compared to moisture content. The protein content of extracted PSC at the 24 h and 48 h extraction time from the skin of silver catfish was lower compared to a study conducted by Kittiphattanabawon et al. [29] on the extracted PSC for 48 h from the cartilage of brownbanded bamboo shark and blacktip shark, with protein contents of 90.91% and 90.18%, respectively.

### 2.3. Solubility

Figure 1 presents the effect of pH ranging from 1 to 10 on the solubility of extracted PSC at 24 and 48 h extraction time from the silver catfish (*Pangasius* sp.) skin. The extracted PSC at 24 and 48 h extraction time showed that the most solubilized (>80%) collagen was at an acidic value (pH 5). Meanwhile, the lowest solubility was noticed at pH 1 and 3, respectively. All the pH levels showed significant differences (*p* < 0.05) except at pH 4 and 7 (*p* > 0.05). The pattern of the obtained solubility indicated that the extracted PSC at 24 h extraction time was increased in solubility with pH extraction increased until pH 5, then slightly decreased at pH 6, and fluctuated until pH 9. Extracted PSC at 48 h extraction time indicated increased solubility from pH 3 until pH 5, then decreasing until pH 9.

At similar optimal pH levels, PSC with both 24 h and 48 h extraction times from silver catfish showed the highest solubility. The solubility was due to the effect of the repulsion properties of collagen molecules above the isoelectric point (pI), which is influenced by the treatment with pepsin [6]. The pepsin treatment significantly impacted the solubility of extracted PSC at 24 h and at 48 h due to the hydrolysis of collagen crosslinks in the telopeptide region cleaving without damaging the triple helix structure, thus achieving higher solubility [30]. According to Matmaroh et al. [27], PSC was the most soluble at a pH range of 2 to 5, with more than 80% relative solubility. The high solubility of extracted PSC at pH 5 agrees with the current study. The PSC at 24 h and 48 h of extraction time showed the highest solubility at pH 5, with more than 80% relative solubility.

Furthermore, PSC extracted from silver catfish skin at 24 and 48 h extraction times had the least solubility at pH levels 1 and 3, respectively. This was attributed to hydrophobic interactions between protein molecules, leading to precipitation and aggregation [31]. In addition, hydrophobic interactions between collagen molecules cause a decrease in the solubility of collagen [32]. However, the results obtained from the current study do not agree with the Nalinanon et al. study, in which the PSC extracted at 24 h and at 48 h from bigeye snapper skin exhibited the lowest solubility at pH 7 [33].

### 2.4. Structural Properties

The structural properties of PSC extracted from the skin of silver catfish (*Pangasius* sp.) at 24 h and 48 h of extraction time are shown in Table 3. The Fourier transform infrared (FTIR) spectra of PSC at 24 and 48 h of extraction time showed the wavenumber peaks of Amides A, I, II, and III. There was no significant difference between PSC at 24 and 48 h extraction time regarding wavelength and vibration mode (*p* > 0.05). In addition, the results of this study were correlated with a functional group (Figure 2), which indicated that there were no discernible differences in the structure of the surfaces of extracted PSC at 24 and 48 h.

The wavenumber of the Amide A band did not show any significant differences in PSC at 24 and 48 h extraction time. The values were observed at 3443.27 ± 0.035 cm^−1^ and 3446.19 ± 7.50 cm^−1^, respectively (*p* > 0.05). The Amide A band is associated with the stretching vibration of N–H bonds, typically occurring within 3400–3440 cm^−1^ [7,28]. The presence of an Amide A band in the extracted PSC at 24 and 48 hrs. suggested that hydrogen bonds are in their protein structures [28]. The current study’s findings agreed with those of Nalinanon et al., who reported an Amide A band of extracted PSC at 24 h on Arabesque greenling (*Pleurogrammus azonus*) skin at 3296 cm^−1^ [34]. In addition, the results also agreed with the study of Veeruraj et al. [23], who reported that an Amide A band of extracted PSC at 48 h on marine eel-fish (*Evenchelys macrura*) was observed at 3395 cm^−1^. A lower Amide A band wavenumber indicates a greater involvement of N–H groups in hydrogen bonding. Therefore, pepsin treatment was not affected by the protein structure of extracted PSC at 24 and 48 h extraction time.

The demonstrated Amide I band indicated no significant difference between PSC at 24 and 48 h extraction time, observed at wavenumbers of 1636.95 ± 0.15 cm^−1^ and 1634.95 ± 3.12 cm^−1^, respectively (*p* > 0.05). This band was commonly linked to stretching vibrations of the C=O carbonyl group along the polypeptide backbone. A helpful indicator for peptide secondary structure is typically found within the frequency range of 1600 to 1700 cm^−1^ [35]. Shanmugam et al. [36] observed that the Amide I band could be shifted to a lower wavenumber due to a reduction in molecular order. Therefore, the current study obtained similar values in the extracted PSC at 24 and 48 h extraction time. The Amide I wavenumber of extracted PSC at 24 h from silver catfish was lower than that extracted at 24 h extraction time from the grass carp’s (*Ctenopharyngodon idella*) skin and swim bladder (1659 cm^−1^ and 1653 cm^−1^) [29]. However, the Amide I wavenumber of PSC extracted from silver catfish at 28 h extraction time was like PSC extracted at 48 h extraction time from the cartilage of blacktip shark (*Carcharhinus limbatus*) and brown-banded bamboo shark (*Chiloscyllium punctatum*) (1634 cm^−1^ and 1633 cm^−1^) [37]. The shift of the amide I band towards a lower wavenumber could be attributed to the partial elimination of telopeptide during the extraction of PSC. This elimination process may cause a loss of lysine and hydroxylysine, which are reactive amino acids [37].

There is no significant difference of Amide II bands between those on PSC extracted at 24 and 48 h, as observed by the wavenumber of 1545.58 ± 0.42 cm^−1^ and 1554.59 ± 8.23 cm^−1^, respectively (*p* > 0.05). This band was generally correlated to the N–H bending vibration with characteristic frequencies ranging between 1500 cm^−1^ to 1600 cm^−1^. Sanden found a correlation between Amide II’s wavenumber and collagen structure’s strength [38]. As a result, the difference in PSC extracted from silver catfish after 24 and 48 h did not affect collagen structure strength. In this current study, the findings were correlated with rheological properties such as thermal denaturation (T_d_) and viscoelastic properties, revealing no significant difference between the PSC at 24 and 48 hr. extraction times. The wavenumber of Amide II on extracted PSC at the 24 h extraction time of silver catfish skin agreed with the finding on extracted PSC for the 24 h extraction time of hybrid *Clarias* sp. skin observed at 1551 cm^−1^ [39]. The Amide II wavenumber of extracted PSC at the 48 h extraction time of silver catfish skin agreed with the finding on extracted PSC at the 48 h extraction time of black tip shark skin, which was observed at 1541 cm^−1^ [22].

The Amide III values also revealed no significant difference between PSC extracted from silver catfish skin after 24 and 48 h at wavenumbers of 1257.70 ± 10.62 cm^−1^ and 1250.23 ± 12.36 cm^−1^, respectively (*p* > 0.05). This band was generally correlated with the N–H bending vibration. The triple helical structure of collagen is confirmed by the presence of Amide III in the sample [40]. Amide III showed the collagen triple-helical structure of extracted PSC at the 24 and 48 h extraction times. As a result, this finding demonstrated that the extracted PSC from silver catfish skin at 24 and 48 h extraction times had no adverse effect on the collagen protein structure. Furthermore, the wavenumber of the amide III and the extracted PSC at 24 h extraction time of silver catfish skin agreed with the study of Wang et al. [41], who found the wavenumber of the Amide III bands of extracted ASC and PSC at 24 h extraction time of deep-sea redfish skin at 1240 cm^−1^ and 1240 cm^−1^, respectively. Additionally, the Amide III wavenumber of the PSC extracted at 48 h of silver catfish skin agreed with the Matmaroh et al. study [27]. They discovered that the extracted PSC and ASC of spotted golden goatfish (*Parupeneus heptacanthus*) scale were at wavenumbers of 1234 cm^−1^ and 1237 cm^−1^, respectively, after a 48 h extraction time. Thus, the prevalence and strength of hydrogen bonds were attributed to the dominant presence of the triple helical structure.

### 2.5. Microstructure Properties

Figure 3 illustrates the silver catfish skin PSC microstructure at 24 and 48 h extraction time. PSC at 24 and 48 h extraction time showed similar structures under a scanning electron microscope (SEM), with similar highly ordered structures. The extraction time of PSC at 24 and 48 h does not affect the collagen microstructure. Both the extracted PSC at 24 and 48 h extraction times was characterized by irregular fibrous networks that contained oversized and irregular pores among the fibrils. The extraction times of PSC at 24 and 48 h did not affect the collagen microstructure. The structure of both collagen showed a flaky and porous structure.

Moreover, it has a porous structure with twisted fibrils connected to the collagen. These results are like those of Wang et al. [42], who examined the Amur sturgeon (*Acipenser schrenckii*) muscle and found that some fibrils were twisted together. Twisted fibrils may have occurred due to the high collagen concentration before self-assembly. The collagen also consists of fibrils that are connected to the flaky collagen. Hence, the collagen connected by fibrils was observed. In addition, the surface of collagen exhibited some partial wrinkles and the presence of surface pores whose size was influenced by the water content during preparation, with a higher water content resulting in larger collagen pores [43]. These probably happen because of dehydration during freeze-drying [23], which also causes wrinkles on the collagen surface [44]. The extracted collagen shows a similar surface morphology to that discussed in Table 3, where no significant difference in the functional group was observed between PSC at 24 and 48 h extraction time.

### 2.6. Rheological Properties

#### 2.6.1. Frequency Sweeps

Figure 4 illustrates the rheological properties of extracted PSC at 24 and 48 h extraction time of silver catfish skin and how the frequency sweep influences them. It showed the complex viscosity (η*) and tan (δ) of collagen solution as a function of frequency (0.01–10 Hz). Figure 4a shows that the PSC at 24 h extraction time has the highest η* compared to the PSC at 48 h extraction time. Both collagen extractions indicated shear-thinning flow behaviour, as evidenced by a nearly linear decrease in η* with increasing frequency (Figure 4a). The behaviour of the collagen solutions for both extractions is regular. Figure 4b shows that the tan (δ) decreases as the frequency increases.

The complex viscosity (η*) of PSC extracted at 24 h showed a higher η* compared to PSC at the 48 h extraction time, which was due to the viscosity of collagen being influenced by inter—and intramolecular hydrogen bonding, in which the difference of viscoelastic led to different amino acid composition, including hydroxyproline content [45]. The extracted PSC at 24 and 48 h extraction time indicated the shear-thinning flow behaviour as η* nearly linear with increasing frequency (Figure 4a). Furthermore, the η* was associated with a higher frequency, resulting in elastic deformation energy. In contrast, the energy of viscous residues was low, indicating that elasticity was the dominant position in collagen liquid [46]. Furthermore, tan (δ) is the G’’/G’ ratio. Yang and Kaufman defined the value of G’ as the elastic energy stored in the system [47]. Meanwhile, the value of G’’ is the energy eliminated through the viscous effects. The lower the tan δ value, the more elastic the material.

The effect of frequency sweep on the rheological properties of extracted PSC at 24 and 48 h extraction time of silver catfish skin agreed with a study by Li et al. [48] on the rheological behaviour of acylated pepsin-solubilized collagen. They discovered that as the frequency increased, the collagen solution’s behaviour changed from elastic to viscous, as evidenced by a continuous decrease in the delta tangent (tan). While Zhang et al. [45] discovered that the loss of tangent (tan) crossed the threshold, the threshold can be observed transitioning from solid-like to liquid-like behaviour. This finding was similar to Moraes et al.’s [49] study on xanthan and guar gums. They found that guar gum dispersion indicated that tan (δ) values are higher than unity at low frequencies but decrease to below unity at higher frequencies. As a result, the decrease in tan (δ) with increasing frequency results in a prequel regime.

#### 2.6.2. Temperature Sweeps

The rheological properties of extracted PSC at 24 and 48 h extraction time of silver catfish skin is depicted in Figure 5 with a temperature sweep from 4 to 50 °C. The collagen solution’s complex viscosity (η*) and tangent (tan) values changed as the temperature increased. First, there were temperature changes for the η* values of extracted PSC at 24 and 48 h extraction time, which gradually decreased in magnitude with a temperature range of 5 °C to 27.5 °C for PSC at the 24 h extraction time and 5 °C to 25 °C at the 48 h extraction time. Then, the η* values of PSC at 24 and 48 h extraction time were sharply decreased in magnitude at a temperature range of 27.5 °C to 32.5 °C for extracted PSC at 24 h extraction time and 25 °C to 32.5 °C for extracted PSC at 48 h extraction time.

The temperature was constant at 32.5 °C until it reached 50 °C for both collagen extractions. Then, tangent (tan) values for extracted PSC at 24 and 48 h extraction times gradually increased in magnitude over a temperature range of 5 °C to 27.5 °C for extracted PSC at 24 h extraction time and 5 °C to 25 °C for extracted PSC at 48 h extraction time. After that, the tan δ value of the extracted PSC at 24 and 48 h extraction time sharply increased in magnitude in a temperature range of 27.5 °C to 32.5 °C for extracted PSC at 24 h extraction time and 25 °C to 32.5 °C for extracted PSC at 48 h extraction time. Finally, the temperature was constant at 32.5 °C until 50 °C for both collagen samples at different extraction times.

The results obtained in this study are due to the tangible changes reflected in the breakdown of the collagen’s triple helix structure into a random coil. This structural alteration may cause bond breakage and stabilize collagen’s secondary structure [50]. A sharp decrease of η* in the magnitude and tan δ parallel the increases (25–32 °C) for both collagen solutions, indicating the fall of the collagen triple helix to a random coil [51]. A decrease in the η* value of PSC at 24 and 48 h extraction time was observed with an increase in temperature. This phenomenon might be due to the increased energy for heat motion of the polypeptide chain as temperature increases. As the temperature increases, the resistance to the segment motion becomes weaker [45]. Furthermore, the denaturation temperature (Td) of the collagen solution under dynamic rheological conditions can be determined as the temperature of η* decreases until 50% of the initial value or at tan reaches the peak value [45,51]. Therefore, the Td values of extracted PSC at 24 h extraction time for two temperatures were observed at 25.5 °C (±0.4 °C) and 29.4 °C (±0.5 °C), respectively; meanwhile, for extracted PSC at 48 h extraction time, the values were 25.3 °C (±1.0 °C) and 30.1 °C (±0.3 °C), respectively. The denaturation temperature (Td) of the extracted PSC at 24 and 48 h extraction time did not show a significant difference (*p* > 0.05).

Regarding dynamic rheological measurements, the T_d_ values of extracted PSC at 24 and 48 h of extraction time agreed with the T_d_ value of a fish-soluble collagen solution measured by Zhang et al. [45] at 29.5 °C (±0.3 °C). However, the extracted PSC at 24 and 48 h extraction times were lower than bovine soluble collagen solution (32.6 ± 0.1 °C) [51]. According to Alves et al. [52], the reduction in η* and a significant rise in tan δ indicate the brittleness and further breakdown of the collagen’s triple helix in the random coil. In addition, Ren et al. [46] reported that the sudden decrease of the crossover frequency at a temperature of 30 °C indicated no significant involvement of the collagen solution, which was close to the denaturation temperature of collagen. This activity may result in tropocollagen molecular movement, specifically by dividing the hydrogen bonds between the CO and NH groups of nearby chains and the triple-helix structure of collagen breakdown, resulting in increased molecular mobility [46].

## 3. Conclusions

In conclusion, the pepsin soluble collagen (PSC) from the skin of silver catfish (*Pangasius* sp.) extracted at different times (24 h and 48 h) and its effect on yield and physicochemical properties determined. PSC at 24 h and 48 h extraction time showed no significant differences in yield. However, the different extraction times influenced the chemical composition of PSC at 24 h and 48 h in terms of moisture, protein, fat, and ash content. Then both collagen extractions showed higher solubility in the acid pH range. Analysis of the functional group wavenumbers revealed that the Amides A, I, II, and III peaks shown as a fingerprint for collagen structure in both samples are in the same range. The scanning electron microscope (SEM) images of both collagen extractions indicate similar morphological structures. The complex viscosity increased exponentially while the loss tangent reduced with frequency. To summarize, PSC extracted after 24 h would be superior regarding chemical properties and extraction time.

## 4. Materials and Methods

### 4.1. Materials

Fresh silver catfish (*Pangasius* sp.) were obtained from the local supplier in Manir of Terengganu state, Malaysia. The fish were kept chilled during transport to the laboratory at FPSM, UMT, for further processing. The chemicals used, including acetic acid and pepsin, were supplied by R & M Marketing, Essex, UK. Analytical-grade chemicals and reagents are used for this study.

### 4.2. Methods

#### 4.2.1. Sample Preparation

Fresh silver catfish were degutted, filleted, and beheaded manually. The silver catfish skins were rinsed with plentiful water to remove contaminants such as blood and soil. The fish skin was chopped into small pieces before homogenization and preserved in a polyethene bag at −20 °C for further use.

#### 4.2.2. Pre-Treatment of Sample

Before extracting the collagen, the pre-treatment (removal of non-collagenous matter and defatting) was conducted. This step aimed to remove unwanted materials and extract collagen in its pure form. Thus, it became free from any other substances that could alter its characteristics. The non-collagenous proteins were removed by soaking each sample in the 0.1 M NaOH solution at 4 °C with a sample-to-solution ratio of 1:8 (*w/v*). The mixture was continuously stirred for six hours, and the NaOH solution was changed every three hours. The treated skins were then washed with ice-cold water until the rinsed water became neutral [12].

The defatting process was done by first removing proteins from the skins; then they were defatted with 10% butyl alcohol for 24 h while being continuously stirred, using the sample-to-solution ratio of 1:20 (*w/v*). The solution was refreshed every 12 h. Finally, the defatted skins were rinsed with 15 volumes of ice-cold water (4–5 °C) [12].

#### 4.2.3. Extraction of Pepsin-Soluble Collagen (PSC)

The pretreated samples were then extracted, followed by the method described by Shanmugam et al. [36], with a slight modification in extraction time. First, the defatted skins were soaked in 0.5 M acetic acid and digested with 1% (*w/w*) pepsin (Sigma-Aldrich, Inc., St. Louis, MO, USA) at a sample-to-solution ratio of 1:20. The resultant mixture was stirred at 4 °C for 24 h. The supernatants of pepsin-solubilized collagen were then centrifuged (150 R, Gyrozen, Gimpo, Republic of Korea) at 10,000× *g* for 30 min at 4 °C for 24 h. Next, the residues were subjected to a second extraction using 0.5 M acetic acid and 1% pepsin for 12 h. The resulting mixture was centrifuged at 10,000× *g* for 30 min at 4 °C. Finally, the supernatants from both extractions were combined, and NaCl was added to achieve a final concentration of 0.7 M, causing the precipitate to form. The supernatant was then centrifuged again at 2500× *g* to obtain the precipitate, and then the precipitate was allowed to freeze-dry. The resultant freeze-dried substance was pepsin-soluble collagen (PSC). A similar procedure was repeated for the collagen extracted within 48 h of extraction time. As a result, the yield of the extracted collagen at different extraction times was calculated as follows:$$\mathrm{Yield}\;\left(\%\right)=\frac{\mathrm{Weight}\;\mathrm{of}\;\mathrm{extracted}\;\mathrm{collagen}\;\left(g\right)\times 100}{\mathrm{Weight}\;\mathrm{of}\;\mathrm{wet}\;\mathrm{skin}\;\left(g\right)}$$

#### 4.2.4. Chemical Composition of Extracted Collagen

The chemical compositions, including moisture, ash, fat, and protein content of the silver catfish skins and extracted collagen, were determined using the AOAC method [53].

#### 4.2.5. Solubility Determination

The effect of pH levels on collagen solubility was determined by the method followed by Kittiphattanabawon et al. with slight modifications [54]. First, the lyophilized collagen was dissolved in the 0.5 M acetic acid with continuous stirring for 12 h to get 3 mg/mL of the final concentration. Next, about 8 mL of sample were transferred into a centrifuge tube, and the pH adjusted its range from 1 to 10 with 6 N NaOH or 6 N HCl; this is followed by making up the volume to 10 mL with distilled water. The solution was stirred at 4 °C for 30 min and then centrifuged (150 R, Gyrozen, Gimpo, Republic of Korea) at 10,000× *g* for 30 min at 4 °C. The protein concentrations in the supernatant were measured according to Lowry’s method [55]. The bovine serum albumin was used as a protein standard. The relative solubility of collagen was calculated as follows:$$\mathrm{Relative}\;\mathrm{solubility}\;\left(\%\right)=\frac{\mathrm{Protein}\;\mathrm{concentration}\;\mathrm{of}\;\mathrm{supernatant}\;\times 100}{\mathrm{The}\;\mathrm{highest}\;\mathrm{protein}\;\mathrm{concentration}}$$

The analysis was conducted in triplicate.

#### 4.2.6. Structural Properties Determination of Extracted Collagen

The extracted collagen’s functional groups were analyzed as Sun et al. [8] described. Approximately 1 mg of lyophilized collagen sample was mixed with dried potassium bromide (KBr) (spectrum pure) at an approximate 1:100 ratio and, under dry conditions, the pellets were prepared. A single scan of infrared spectra (iS 10, Thermo Fisher Scientific, Waltham, MA, USA) was obtained at the resolution of 1 cm^−1^ and 4000–400 cm^−1^ range of frequency. In addition, the OMNIC 8.0 software was used to analyze the peak wavelength. These analyses were conducted in triplicate.

#### 4.2.7. Microstructure Determination of Extracted Collagen

The microstructural characteristics of extracted collagen were observed with scanning electron microscopy (SEM) (S-4300SE, Hitachi, Tokyo, Japan) and an accelerating voltage of 5.0 Kv. The collagen powder was mounted on 5 mm × 12.5 mm dimensional aluminium cylinder stubs and sputter-coated using a fine auto coater (JFC 1600, Tokyo, Japan). The samples were observed at 500× magnification in a superficial layer, and analysis was done in triplicate [12].

#### 4.2.8. Rheological Characterisation of Collagen Solution

The rheological characteristics of the extracted collagen were analyzed, followed by Lai et al. [51]. Then, with the help of a Discovery HR2 Rheometer (TA instrument, New Castle, DE, USA), oscillatory rheological parameters were performed using stainless steel cone/plate geometry (4° cone angle and 40 mm cone diameter) on the gap set at 150 µm. A Peltier temperature controller was used to achieve temperature control. The range was 4 to 50 °C, with an accuracy of ±0.1 °C.

Frequency sweeps determined the viscoelastic behaviour of collagen at different extraction times. The dynamic frequency sweeps for the PSC at 24 and 48 h extraction time were performed using the frequency parameters with the range of 0.01 to 10 Hz at 25 °C with a constant strain of 5%. The analysis was conducted in triplicate.

Temperature sweeps determined the viscoelastic behaviour of the collagen at different extraction times. Within a linear range, the dynamic temperature sweeps were carried out at a frequency of 1 Hz and a constant strain of 5%. The collagen solution was prepared by adding 3 g of collagen powder to 2 mL of distilled water. The collagen solution was heated at a rate of 0.5 °C/min from 4 to 50 °C. This analysis was completed in triplicated samples.

### 4.3. Statistical Analysis

The data were reported as mean ± standard deviation values of three repeats. A *t*-test was used for statistical analysis, and a probability value of *p* < 0.05 was considered a significant difference. All computations were performed using Minitab Software (Version 14, 2008).

## Figures and Tables

**Figure 1 gels-09-00300-f001:**
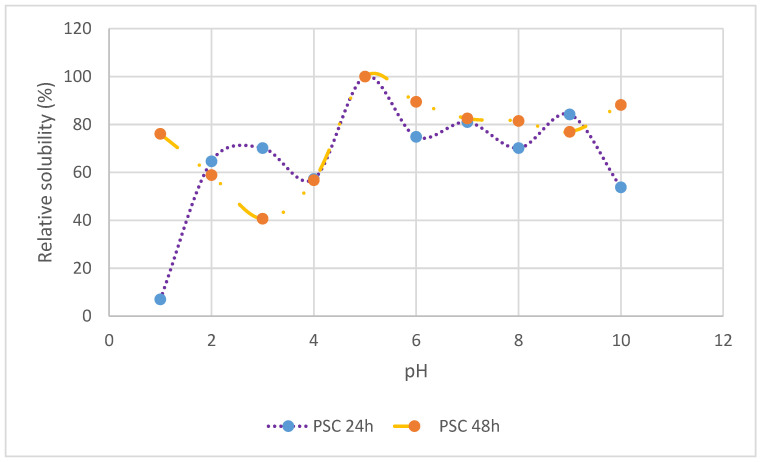
Solubility of PSC extracted from skin of silver catfish (*Pangasius* sp.) at 24 h and 48 h extraction time in the pH range of 1 to 10.

**Figure 2 gels-09-00300-f002:**
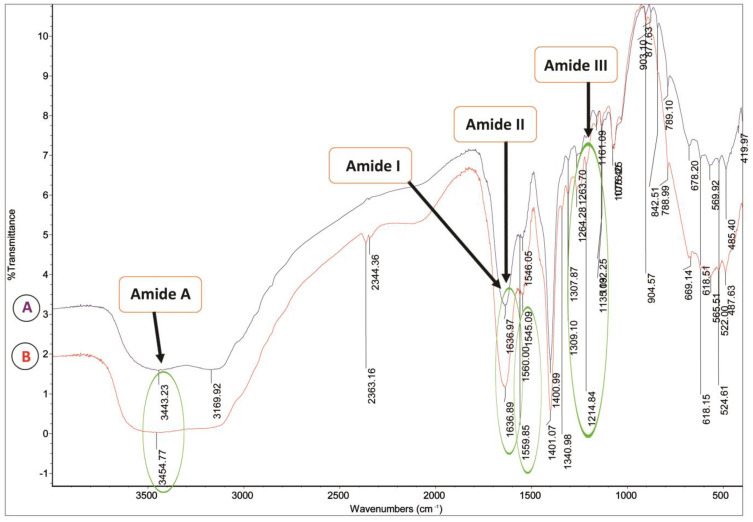
Fourier Transform Infrared Spectroscopy (FTIR) spectra for PSC from skin of silver catfish (*Pangasius* sp.) at (**A**) 24 h and (**B**) 48 h shows Amide A, I, II, and III functional groups.

**Figure 3 gels-09-00300-f003:**
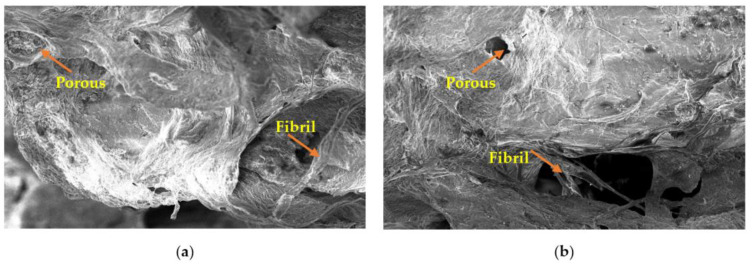
Scanning electron microscopic (SEM) images of PSC at 24 h (**a**) and 48 h (**b**) shows fibril and porous like structure.

**Figure 4 gels-09-00300-f004:**
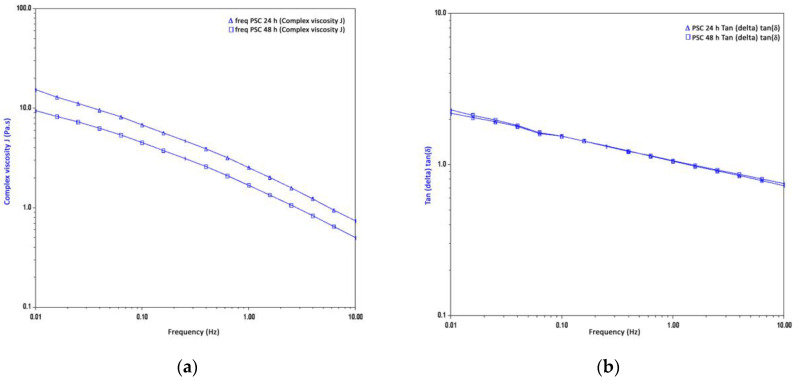
The frequencies of (**a**) complex viscosity (η*) and (**b**) loss tangent (tan δ) of collagen solution of extracted PSC at 24 and 48 h extraction time of silver catfish.

**Figure 5 gels-09-00300-f005:**
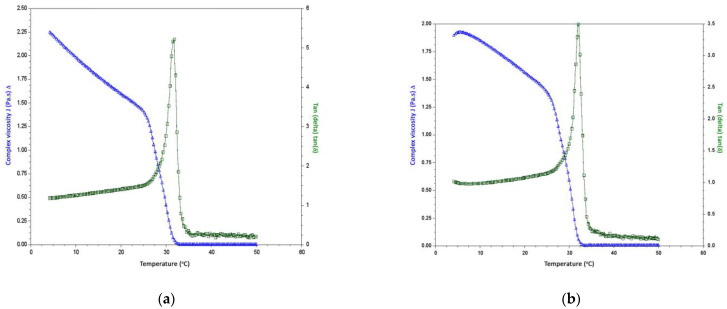
Temperature dependence of complex viscosity (η*) and tangent (tan δ) of silver catfish collagen at (**a**) 24 h and (**b**) 48 h extraction time.

**Table 1 gels-09-00300-t001:** The yield of extracted silver catfish skin collagen at different extraction times.

Extraction Time (h)	Yield (%)
PSC 24	23.64 ± 2.86 ^a^
PSC 48	26.43 ± 2.48 ^a^

The data are presented in mean ± standard deviation from three replications (*n* = 3). Similar letters (a) showed no significant difference (*p* > 0.05) within columns.

**Table 2 gels-09-00300-t002:** Chemical composition of pepsin soluble collagen (PSC) 24 h and 48 h extraction time.

Sample	Chemical Composition (%)			
	Moisture	Ash	Fat	Protein
PSC 24 h	93.15 ± 0.48 ^a^	21.86 ± 0.88 ^b^	2.95 ± 0.31 ^a^	53.22 ± 2.34 ^a^
PSC 48 h	89.93 ± 0.35 ^b^	26.69 ± 1.68 ^a^	2.73 ± 0.81 ^b^	45.18 ± 1.24 ^b^

The data are presented in mean ± standard deviation from three replications (*n* = 3). Different letter (a,b) shows significant difference (*p* < 0.05) within column.

**Table 3 gels-09-00300-t003:** Structural properties of extracted PSC at 24 and 48 h different extraction time.

Functional Group	Mode of Vibration	Wavelength (cm^−1^)	
		24 h	48 h
Amide A	N–H stretching	3443.27 ± 0.035 ^a^	3446.19 ± 7.50 ^a^
Amide I	C=O stretching	1636.95 ± 0.15 ^a^	1634.95 ± 3.12 ^a^
Amide II	N–H bending	1545.58 ± 0.42 ^a^	1554.59 ± 8.23 ^a^
Amide III	N–H bending	1257.70 ± 10.62 ^a^	1250.23 ± 12.36 ^a^

The data are presented in mean ± standard deviation from three replications (n = 3). Similar letters (a) show no significant difference (*p* > 0.05) within rows.

## Data Availability

The data presented in this study are available on request from the corresponding author.

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
