# Peer review of "Effect of Extraction Time on the Extractability and Physicochemical Properties of Pepsin—Soluble Collagen (PCS) from the Skin of Silver Catfish (*Pangasius* sp.)"

_gels, 2023, doi:10.3390/gels9040300_

Round 1
Reviewer 1 Report
The manuscript is written well. A few corrections are recommended.
Lines 29-30. Construction of sentence needs to be modified. It is precious because it can be used for treatment of the diseases?
Line 39. Promising results on what?
Author Response
|
Reviewer-1 |
|
The manuscript is written well. A few corrections are recommended. |
|
Lines 29-30. Construction of sentence needs to be modified. It is precious because it can be used for treatment of the diseases?
The sentence has been revised (Lines 30-33). |
|
Line 39. Promising results on what?
The sentence has been revised (Lines 41-42). |

Reviewer 2 Report
This manuscript studies the effect of extraction time on the extractability and physicochemical properties of collagen extracted from the skin fish of silver catfish. The author found that as the extract time increases 2 times, the extractability only slightly increases. This paper also found that the extraction time does not influence the extractability and physiochemical of the pepsin-soluble collagen (PSC). However, there are some questions that need to be answered and revised. Here are some comments:
- Please add more relevant research on how the extraction time can influence the properties of the collagen in the introduction section. Previous research found no significant effect on the extraction time on the properties of the extracted collagen. What makes this current research differ from the previous study? Please explain in detail to show the novelty of this current research.
- Why did the authors investigate only two extraction times? Please also explain in detail whether the short range of extraction time might result in almost similar yields since the enzymatic hydrolysis will have optimum conditions.
- It would be advisable to include the FTIR spectra of both PSC extracts (24 h and 48 h extraction). The functional group can be identified by FTIR, but not the molecular structure. It is advisable to include UV-Vis analysis to confirm the FTIR data.
- It is important to discuss also the chemical properties of both extracts PSC such as chemical composition, pH, and amino acids composition, instead of the physical properties differences.
Author Response
|
Reviewer-2 |
|
This manuscript studies the effect of extraction time on the extractability and physicochemical properties of collagen extracted from the skin fish of silver catfish. The author found that as the extract time increases 2 times, the extractability only slightly increases. This paper also found that the extraction time does not influence the extractability and physiochemical of the pepsin-soluble collagen (PSC). However, there are some questions that need to be answered and revised. Here are some comments: |
|
Please add more relevant research on how the extraction time can influence the properties of the collagen in the introduction section. Previous research found no significant effect on the extraction time on the properties of the extracted collagen. What makes this current research differ from the previous study? Please explain in detail to show the novelty of this current research.
The influence of extraction time on various properties has been revised and added in introduction part. |
|
Why did the authors investigate only two extraction times? Please also explain in detail whether the short range of extraction time might result in almost similar yields since the enzymatic hydrolysis will have optimum conditions.
According to previous studies on extraction of collagen, these two extraction times referred to be most optimum conditions for marine sourced collagen extraction. |
|
It would be advisable to include the FTIR spectra of both PSC extracts (24 h and 48 h extraction). The functional group can be identified by FTIR, but not the molecular structure. It is advisable to include UV-Vis analysis to confirm the FTIR data.
The sentence has been revised and the FTIR spectra of both PSC extracts was added as figure 2. |
|
It is important to discuss also the chemical properties of both extracts PSC such as chemical composition, pH, and amino acids composition, instead of the physical properties differences.
The chemical composition of PSC at 24h and 48h has been included in results and discussion part and data was represented in table 2. |

Reviewer 3 Report
1. Application of enzymes and immobilized enzymes is a big field; I recommend to cite some recent reviews which cover this research field. For example,
Immobilization of Penaeus merguiensis alkaline phosphatase on gold nanorods for heavy metal detection (https://www.sciencedirect.com/science/article/pii/S0147651316304274)
2. Materials and Methods: Quite a good number of experiments were presented. However, they have been split into so many sub-sections. Co-related subsections could be merged. Besides, many of the experimental procedures lack details that should be provided/included.
3. Results and Discussion: Overall, adequate Data/information has been reported/presented. However, there needs scientific explanation/reasoning (e.g. why/how) of the Data which mostly lacks.
4. Conclusions: Should summarize the significant findings of ALL THE MAJOR STUDIES, and thus needs to be extended further within the scopes.
5. Ideally, the figure captions should be informative and representative. Currently, most of the captions are just the name of the figure(s) and thus require further extension within the limit.
Author Response
|
Reviewer-3 |
|
1. Application of enzymes and immobilized enzymes is a big field; I recommend to cite some recent reviews which cover this research field. For example, Immobilization of Penaeus merguiensis alkaline phosphatase on gold nanorods for heavy metal detection (https://www.sciencedirect.com/science/article/pii/S0147651316304274)
This article not suitable to our research, hence, we are unable to include this citation in this manuscript. |
|
2. Materials and Methods: Quite a good number of experiments were presented. However, they have been split into so many sub-sections. Co-related subsections could be merged. Besides, many of the experimental procedures lack details that should be provided/included.
The materials and methods part has been revised as per reviewer suggestion and co-related subsections was merged. |
|
3. Results and Discussion: Overall, adequate Data/information has been reported/presented. However, there needs scientific explanation/reasoning (e.g. why/how) of the Data which mostly lacks.
The sufficient scientific explanations are given in results and discussion part with recent citations to address the why/how our results justified with previous studies. |
|
4. Conclusions: Should summarize the significant findings of ALL THE MAJOR STUDIES, and thus needs to be extended further within the scopes.
The conclusion part has been revised and improved with including all major finding of this study. |
|
5. Ideally, the figure captions should be informative and representative. Currently, most of the captions are just the name of the figure(s) and thus require further extension within the limit.
The captions for figures have been revised according to reviewer comments. |

Round 2
Reviewer 2 Report
I would suggest drawing Figures 3 and 4 as the previous version (including the X-axis and unit label as well.